

# An unfolding method based on conditional invertible neural networks (cINN) using iterative training

Mathias Backes[1⋆], Anja Butter[2,3†], Monica Dunford[1‡] and Bogdan Malaescu[2∘]

**1** Kirchhoff-Institut für Physik, Universität Heidelberg, Germany
**2** LPNHE, Sorbonne Université, Université Paris Cité, CNRS/IN2P3, Paris, France
**3** Institut für Theoretische Physik, Universität Heidelberg, Germany

⋆ mathias.backes@kip.uni-heidelberg.de , † anja.butter@lpnhe.in2p3.fr ,
‡ monica.dunford@kip.uni-heidelberg.de , ∘ malaescu@in2p3.fr

## Abstract

The unfolding of detector effects is crucial for the comparison of data to theory predictions. While traditional methods are limited to representing the data in a low number of dimensions, machine learning has enabled new unfolding techniques while retaining the full dimensionality. Generative networks like invertible neural networks (INN) enable a probabilistic unfolding, which map individual data events to their corresponding unfolded probability distribution. The accuracy of such methods is however limited by how well simulated training samples model the actual data that is unfolded. We introduce the iterative conditional INN (IcINN) for unfolding that adjusts for deviations between simulated training samples and data. The IcINN unfolding is first validated on toy data and then applied to pseudo-data for the $pp \to Z\gamma\gamma$ process.


# 1  Introduction

To test various theories of physics, we either have to include detector effects in simulations or correct for these effects in experimental data. This procedure of correcting or 'unfolding' the data for imperfect detection efficiencies and resolutions (in the sense of an expected distribution) involves a precise knowledge of the detector's response to the phenomena being measured. To further complicate matters, with classical unfolding methods, this inversion problem is only tangible if the data is represented in a reduced number of dimensions, for example, the energy distribution of all interacting particles entering the detector. Therefore the method can not take into account the dependencies of the response function on hidden variables (i.e. that are not unfolded), which may induce a bias in the unfolding result.

In particle physics, where we enjoy the benefits of excellent first-principle Lagrangian-based event generators and extensive simulations of the detector, multiple unfolding methods have been developed to minimize any dependence of the unfolded data on an imperfect model of the truth distributions. These traditional unfolding techniques employ binned data distributions and simulation-based transfer matrices to describe the connection between the truth- and the reconstructed-level quantities.[1] However, a direct inversion of the transfer matrix can induce large variances in the unfolded distributions (see e.g. Ref. [3]). Alternative methods based on a Singular Value Decomposition (SVD) of the transfer matrix [4] or on a least square fit in TUnfold [5], together with a Tikhonov regularization, aim to reduce this statistical variance at the price of inducing some amount of bias in the unfolded distribution.

To overcome these issues, some unfolding methods employ iterations to reduce the bias caused by differences between data and the simulation for the distributions of the observable(s) of interest. Early methods in this direction (see Ref. [6] and references therein) were typically exemplified for some analytic folding functions, but the methodology described therein can be readily interpreted in the discrete matrix-based unfolding case. Such iterative matrix-based methods are discussed in Refs. [7–16] and references therein.

During the last decades, the need for binning-free unfolding became apparent in several applications [17]. Various methods based on e.g. an iterative weighting of the Monte Carlo events [18], or the migrations of the true quantity, using the concept of energy (i.e. minimizing a test statistics in a series of random migration steps) [19,20] have been developed. Further tremendous improvements in the areas of binned and unbinned unfolding have been achieved using genetic algorithms [21,22]; approaches to unfolding problems involving machine learning concepts, including a training sample, a validation procedure and boosting [23]; Neural Networks [24–26], also in conjunction with modified Least Square methods [27], performing a hyperparameter space scan to find the best network geometry [28], employing iterations [29–32]; as well as fcGAN and cINN techniques [33–35, 37]. In this paper we present the implementation of a new event-by-event iterative cINN-based unfolding approach.

# 2  Iterative cINN unfolding

Let $f_{\text{true}}(x)$ be the true underlying function that we want to measure. Instead of measuring $f_{\text{true}}(x)$ directly, we can only observe a measured distribution of events $g(y)$, which is the result of convoluting the true function with a response function $r(y|x)$.

$$g(y) = \int r(y|x) f_{\text{true}}(x) \, \mathrm{d}x \ . \tag{1}$$

---

[1]Many of these methods are readily available in tools like RooUnfold [1] or RooFitUnfold [2].

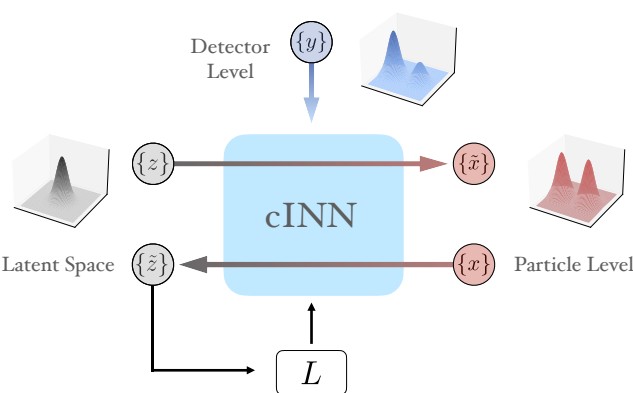

Figure 1: Structure of the conditional INN. Random numbers $\{z\}$ are mapped to particle-level events $\{x\}$ under the condition of a detector-level event $\{y\}$. The loss $L$ follows Eq. (9), a tilde indicates a cINN-generated event.

The response function captures the resolution and efficiency of the detector, implying that it is normalized to one if and only if the detection efficiency is perfect. In this work, we define the truth function (so-called particle level) to include effects from parton showering and hadronization effects. Therefore, $r(y|x)$ describes exclusively resolution effects and can hence be understood as the likelihood function $p(y|x)$ with

$$1 = \int p(y|x)\, \mathrm{d}y \; . \tag{2}$$

Starting from this setting, the goal of unfolding is to obtain the best estimate for $f_{\text{true}}$. Formally the true distribution can be obtained from the measured distribution via a pseudo-inversion

$$f_{\text{true}}(x) = \int p(x|y)g(y)\, \mathrm{d}y \; , \tag{3}$$

where $p(x|y)$ is the posterior, i.e. the conditional probability density at truth level given a measured event $y$. In practice, likelihood and posterior are linked to each other via Bayes' Theorem

$$p(x|y) = \frac{p(y|x)p(x)}{p(y)} \; , \tag{4}$$

given the normalized distribution over data $p(y)$, commonly referred to as evidence, and the prior distribution $p(x)$ of the underlying events. The dilemma of inverse problems arises in that to obtain the posterior, we need to insert the true underlying distribution $p(x)$ in the first place. Several methods, such as iterative Bayesian unfolding have been developed to overcome this limitation. More recently the focus has been on machine learning based unbinned approaches which are scalable to higher dimensions, using either classifier based reweighting techniques or generative models, which can learn the posterior directly. In this work we present a method to combine the advantages of iterative methods with unbinned generative networks.

## 2.1 cINN unfolding

Unfolding methods based on generative networks learn the probability distribution of events at particle level conditioned on detector-level information. The heart of the method consists of a generative network that encodes the posterior $p(x|y)$. It is implemented as a normalizing

flow which have been established for the use of bijective mappings in the machine learning landscape. These networks induce a bijective mapping between a simple, tractable distribution $p_z(z)$ - the latent distribution - and a complex target distribution $p_x(x)$. This property enables the computation of the Jacobian of the mapping for every point of the latent space.

$$p_x(x) = p_z(z)\left|\frac{\partial z}{\partial x}\right| = p_z(z)J_{\text{INN}}. \tag{5}$$

Invertible Neural Networks (INN) are normalizing flows for which the computation of the inverse direction is computationally cheap, i.e. it encodes both mapping directions simultaneously into its network parameters. Studies have demonstrated that invertible networks are particularly well suited to learn the event distributions [36] and hence posterior [37].

A graphical representation of the conditional INN (cINN) [35] used for this result can be found in Fig. 1. In this set-up, an $n$-dimensional random noise, $z$, is mapped onto a distribution of particle-level events, $x$, which also has $n$ inner degrees of freedom. This mapping is bidirectional, thereby a particle-level event, $\tilde{x}$, can be generated by the cINN and afterwards mapped back to the latent space variable, $r$. Detector-level events, $y$, act as a condition for the INN mapping, i.e. the random noise $z$ is linked to particle-level events $x$ under the condition $y$. The bidirectional nature of the mapping is also kept in the conditional approach, which is essential for the evaluation of the loss function [37]. During training, paired samples of detector- and particle-level events are passed through the network to the latent space. Once the training has converged we can sample from the latent space under the condition of a specific detector-level event to generate a distribution of particle-level events, preserving the statistical nature of the unfolding. Without the conditionality the trained network would simply become an event generator which would reproduce the underlying full distribution of particle-level events in the training data set.

The correct calibration of the particle-level distributions is guaranteed through the maximization of the likelihood of the network parameters $p(\theta|x, y)$. The loss function of the cINN encodes therefore the negative log likelihood

$$\mathcal{L} = -\langle \log p(\theta|x, y)\rangle_{x\sim f, y\sim g} \tag{6}$$

$$= -\langle \log p(x|\theta, y)\rangle_{x\sim f, y\sim g} - \langle \log p(\theta|y)\rangle_{y\sim g} + \langle \log p(x|y)\rangle_{x\sim f, y\sim g} \tag{7}$$

$$= -\langle \log p(x|\theta, y)\rangle_{x\sim f, y\sim g} - \lambda\theta^2 + \text{const.} \tag{8}$$

$$= -\langle \log p(z(x)|\theta, y)\rangle_{x\sim f, y\sim g} - \langle \log\left|\frac{\mathrm{d}z}{\mathrm{d}x}\right|\rangle_{x\sim f, y\sim g} - \lambda\theta^2 + \text{const.} \tag{9}$$

In the first line we apply Bayes' theorem to express the posterior on the network parameters in terms of the likelihood of the training data, a prior on the network parameters, and the evidence of the data, which is independent of the trainable network weights and hence irrelevant for the training. Under the assumption of a Gaussian distribution of the network parameters, the prior is equivalent to an $L_2$ regularization and becomes $\lambda\theta^2$ where $\lambda$ encodes the width of the Gaussian prior or the strength of the regularization. In the last line we apply the change of variable formula for a bijective mapping from Eq. (5). The assumption of Gaussian latent space only enters in the definition of $p(z(x))$. [37]

## 2.2 Iterative approach

While the cINN is able to learn a posterior distribution $p(x|y)$, the learned expression will depend on the prior $p(x)$ encoded in the training data. To reduce any biases due to the simulation used in the training, we propose an iterative cINN unfolding. The algorithm is sketched in Fig. 2.

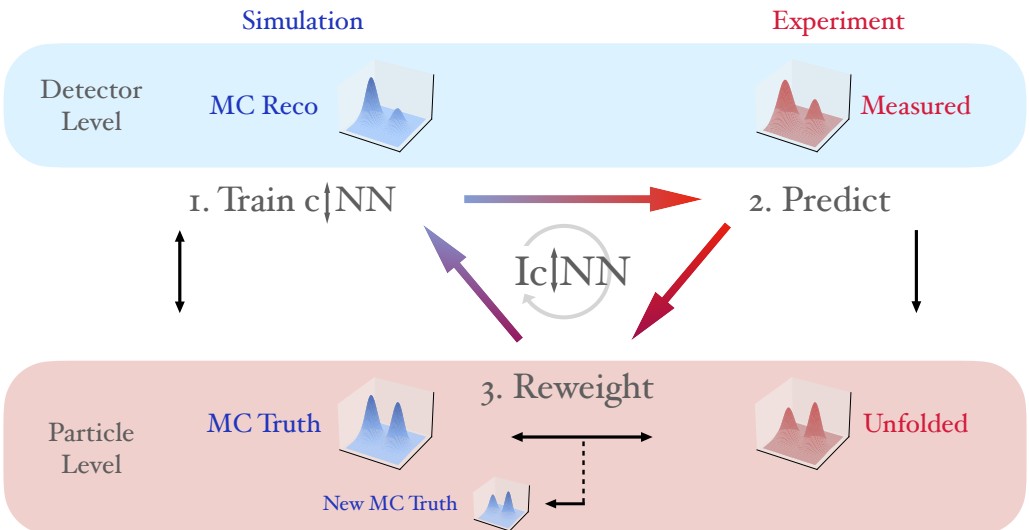

Figure 2: Illustration of the iterative cINN unfolding algorithm. In a first step the regular training of the cINN on the current Monte Carlo Data is performed. As a second step the cINN unfolds the experimentally measured distribution. In a third step the Monte Carlo simulation is reweighted to match the unfolded distribution on Particle Level. This procedure is iterated, always with a modified Monte Carlo Simulation.

- The *first two steps* are identical to the standard cINN setup: we first train the cINN on our simulated data and apply it then to our measured data, i.e. we sample $z$ in the latent space under the condition of a measured event $y$ to obtain unfolded distribution $f_{u,i}(x)$ starting with $i = 0$ for the first iteration.

- In a *third step* we train a classifier to learn the ratio between the phase space densities of the unfolded distribution and the truth-level prior distribution. We then *reweight the simulation on particle-level* to match the unfolded distribution $f_{u,i}(x)$. Since each event of the simulation on particle-level is connected to one event of the simulation on detector-level the event weights can be transferred from particle to detector level.

- We then repeat all steps of training-unfolding-reweighting with the new reweighted simulation until the algorithm has converged.

The effect of this iterative procedure is that we include more and more information of the measured data into our simulation and thereby improve our unfolding result.

On the technical side we find that it is computationally more efficient, if the cINN of the previous iteration is used as a new starting point. To train the cINN on the weighted Monte Carlo, we modify the loss function of the cINN (see Eq. (9)) to be trainable on events with weight $w(x)$ [38, 42]

$$\mathcal{L} = -\langle w(x) \log p(\theta|x,y)\rangle_{x \sim f, y \sim g}. \tag{10}$$

The number of iterations is set to obtain a good balance between the bias towards the Monte Carlo and the statistical uncertainties of the unfolded distribution.

This iterative algorithm allows to minimize potential biases caused by data - Monte Carlo shape differences, while maintaining the advantage of performing a probabilistic unfolding

of all objects for each individual data event. The combination of these features offers an important potential for better interpretability of the unfolded results.

## 3  1D toy example

We start by constructing a analytically solvable Gaussian toy model in order to demonstrate the algorithm. To illustrate the limitations of the cINN unfolding, we construct a challenging scenario, that highlights the prior dependence of non-iterative methods. This means we implement a large difference between the Gaussian truth-level distributions in data and Monte Carlo. In addition, we include significant detector response effects corresponding to a systematic coherent shift and a Gaussian smearing.

Large data to MC differences can be implemented by choosing significantly different mean values for data truth and MC truth distribution. In this example we choose the Monte Carlo distribution at truth-level to be a Gaussian with $f_{\text{MC}}(x) = G(x; \mu_{MC,t} = 4, \sigma_{MC,t} = 4)$. The truth-level data distribution is parameterized as $f_{\text{Data}}(x) = G(x; \mu_{\text{Data,t}} = 10, \sigma_{\text{Data,t}} = 3.8)$.

The detector response function is given by the analytic expression

$$p(y|x) = G(y; (x + \mu_{\text{smear}}), \sigma_{\text{smear}}) = \frac{1}{\sqrt{2\pi \sigma_{\text{smear}}^2}} \exp\left(-\frac{(y - (x + \mu_{\text{smear}}))^2}{2\sigma_{\text{smear}}^2}\right), \quad (11)$$

with $\mu_{\text{smear}} = -6$, $\sigma_{\text{smear}} = 3$.

The respective detector level distributions can be calculated using Eq. (1). The resulting convolution of two Gaussian functions is also a Gaussian function with a mean value and variance equal to the sum of the means and variances of the convoluted Gaussians. We refer to the Gaussian parameters of the distribution $g_{\text{MC}}(y)$ after the convolution (i.e. at reconstructed-level) in Monte Carlo as $\mu_{\text{MC,r}}$ and $\sigma_{\text{MC,r}}$, while the corresponding distribution of measured data $g_{\text{Data}}(y)$ is described by the parameters $\mu_{\text{Data,r}}$ and $\sigma_{\text{Data,r}}$. The complete set of binned distributions is shown on the left side of Fig. 3.

We can now calculate the analytic expectation for an unfolding result after each iteration to compare it to the iterative cINN algorithm. For this we use Bayes theorem (Eq. (4)) to

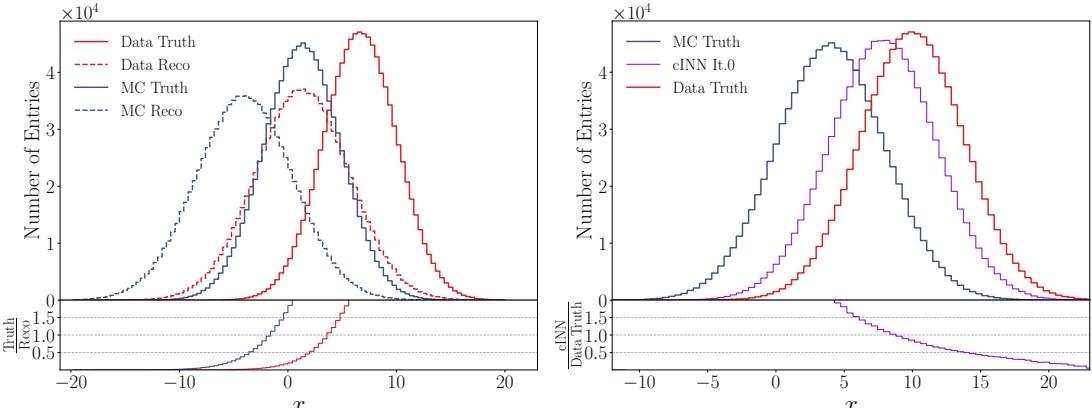

Figure 3: Gaussian toy example used to demonstrate the IcINN algorithm. The left image shows all relevant distributions of the model: the data truth (red, solid) the data reco (red, dashed), the MC truth (blue, solid) and the MC reco (blue, dashed), each with $10^6$ sampled events. On the right the cINN unfolding is applied to the model; the resulting unfolded distribution (purple, solid) is biased towards the MC Truth.

calculate the pseudo-inverted function $p(x|y)$. We start by assuming the Monte Carlo truth $f_{\text{MC,t}}(x)$ as a prior for the unfolded distribution

$$p(x|y) = \frac{p(y|x) f_{\text{MC}}(x)}{g_{\text{MC}}(y)} = \frac{p(y|x) G(x; \mu_{\text{MC,t}}, \sigma_{\text{MC,t}})}{G(y; \mu_{\text{MC,r}}, \sigma_{\text{MC,r}})} . \tag{12}$$

The unfolded distribution can now be calculated according to Eq. (3). By using the previous result for $p(x|y)$ and the experimentally measured distribution $g_{\text{Data}}(y)$ we obtain the first unfolding result, namely

$$f_{u,0}(x) = \int p(x|y) g_{\text{Data}}(y) \, dy \tag{13}$$

$$= \int p(x|y) G(y; \mu_{\text{Data,r}}, \sigma_{\text{Data,r}}) \, dy \tag{14}$$

$$= G(x; \mu_{u,0}, \sigma_{u,0}), \tag{15}$$

with

$$\mu_{u,0} = \frac{(\mu_{\text{Data,r}} - \mu_{\text{smear}}) \sigma_{\text{MC,t}}^2 + \mu_{\text{MC,t}} \sigma_{\text{smear}}^2}{\sigma_{\text{MC,t}}^2 + \sigma_{\text{smear}}^2}, \tag{16}$$

$$\sigma_{u,0} = \frac{\sigma_{\text{MC,t}} \sqrt{\sigma_{\text{MC,t}}^2 \sigma_{\text{Data,r}}^2 + \sigma_{\text{MC,t}}^2 \sigma_{\text{smear}}^2 + \sigma_{\text{smear}}^4}}{\sigma_{\text{MC,t}}^2 + \sigma_{\text{smear}}^2} . \tag{17}$$

As expected, we observe that the unfolded distribution in Fig. 3 deviates from the data truth, since it is biased towards the MC truth. In order to mitigate this bias, in the next step we use the unfolded distribution as an updated prior for the Monte Carlo truth distribution.

This is everything we need for an analytic prediction of the unfolded distribution after each iteration. To obtain the parameters of the distribution after the second iteration we simply re-calculate Eq. (17) replacing $\{\mu_{\text{MC,t}}, \sigma_{\text{MC,t}}\}$ with $\{\mu_{u,0}, \sigma_{u,0}\}$. If the cINN as well as the classifier train perfectly, the obtained unfolded distribution and the analytical result should be identical after each iteration.

Now we are ready to apply the iterative cINN unfolding on this toy example. We employ a cINN with cubic-spline blocks [39] to learn the posterior. For the classifier we use a standard fully connected neural network with a sigmoid activation function for the output node. To improve the classifier training, ADAM [40] and a one-cyclic learning rate scheduler are implemented. In order to build the network structure the framework FrEIA [41] is used. The hyperparameters of the cINN and the classifier are given in Table 1 in the Appendix A.

Since we have full control over the analytic solution which corresponds to an approximately perfect training, it is possible to cross-check the unfolding result after each iteration with the analytic prediction. The result of the IcINN unfolding is displayed on the left in Fig. 4. The results have been obtained using 1000 unfolding steps for each event. The unfolded distribution of each iteration (solid lines) is very close to the respective analytic prediction (dashed line). It is also clearly visible that the bias towards the Monte Carlo simulation is iteratively reduced.

In order to ensure that the networks are converging, we have performed multiple closure-tests for classifier and cINN. This includes tests that the reweigthing of the Monte Carlo simulation reproduces the unfolded distribution and that the unfolding of the MC data reproduces the MC truth. We can hence conclude, that the algorithm is working as expected.

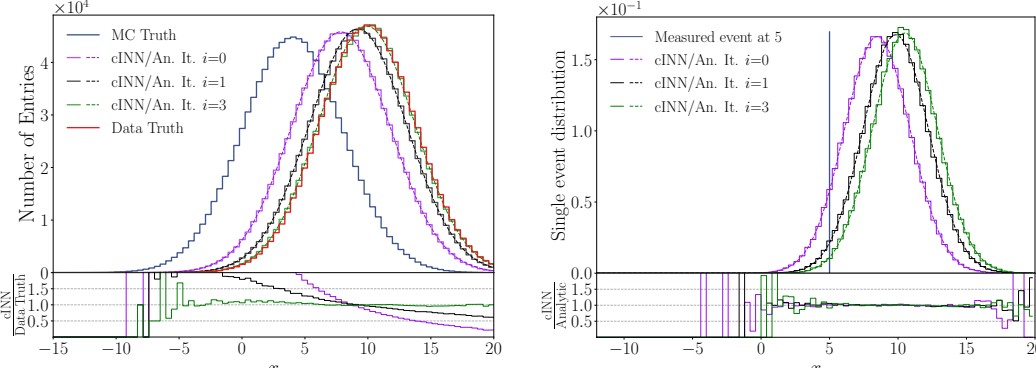

Figure 4: Iterative unfolding results for the one-dimensional toy model. On the left in the upper part we show the MC and data truth as well as the unfolding result in each iteration (solid lines) together with its analytic prediction (dashed line). In the lower part we show the ratio of the cINN with the data truth; it is clearly visible that the bias towards the MC is iteratively reduced. On the right we show the unfolded distribution for a single event at $y_m = 5$. Again the result after each iteration (solid histogram lines) is very close to its analytic prediction (dashed lines).

## 3.1 Single event distribution

One important feature of the cINN unfolding is the possibility to predict distributions for single data events, a property preserved in the iterative approach. In the context of our toy model we can predict analytically how the single event unfolded distribution should look like. The contribution to the data distribution from a single measured event can be expressed with a delta distribution

$$g_{\text{Meas}}(y) = \delta(y - y_m). \tag{18}$$

Plugging the delta distribution into the first part of Eq. (15), we obtain as an unfolded distribution a Gaussian distribution with mean and variance

$$\mu_{\text{single}} = \frac{\sigma_{\text{smear}}^2 \mu_{\text{MC,t}} - \sigma_{\text{MC,t}}^2 (\mu_{\text{smear}} - y_m)}{\sigma_{\text{smear}}^2 + \sigma_{\text{MC,t}}^2}, \qquad \sigma_{\text{single}}^2 = \frac{\sigma_{\text{smear}}^2 \sigma_{\text{MC,t}}^2}{\sigma_{\text{smear}}^2 + \sigma_{\text{MC,t}}^2}. \tag{19}$$

This result can also be derived directly from Eq. (17) by modifying $\mu_{\text{Data,r}} \to y_m$ and $\sigma_{\text{Data}} \to 0$. In the right side of Fig. 4 we show the single event unfolding result together with its analytic prediction in each iteration. The event is unfolded 10 000 times to obtain a smooth distribution. Again, the analytical prediction and the IcINN are in excellent agreement. Hence the IcINN learns correct single event unfolded distributions.

## 3.2 Uncertainties and correlations

In view of a further use of the unfolded distributions in physics studies, the information on the nominal unfolding result has to be complemented with the corresponding statistical uncertainties and their correlations between different phase-space regions. Indeed, such uncertainties have multiple components originating from the finite size of the input data and MC samples, as well as from the cINN training process.[2] These statistical uncertainties and their correlations are visualized here by projecting the unfolding result into a histogram with ad-hoc binning

---

[2]Note that the following procedure does not include systematic uncertainties that can be induced for instance by insufficient expressivity of networks.

and evaluating the corresponding covariance and correlation matrices. This evaluation is performed using a bootstrap method [43], which employs a series of pseudo-experiments where the weight of each data and/or MC event is fluctuated according to a Poisson distribution of mean one. Before we can apply the bootstrap method we have to evaluate the underlying

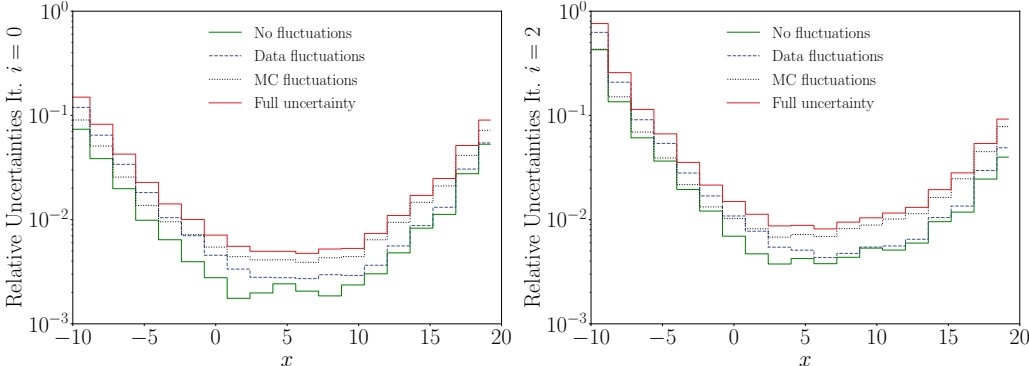

Figure 5: Relative statistical uncertainties of the IcINN before any reweighting, i.e. for iteration $i = 0$ (left), and with two reweightings, i.e. for iteration $i = 2$ (right), evaluated with no fluctuations for the input distributions (green histogram), with fluctuations from data (blue dashed line), MC (black dotted line) and total (red histogram). When deriving these uncertainties, each event is unfolded 30 times and a number of $N_{\text{toys}} = 400$ bootstrap replicas is used.

noise level that originates simply from training and evaluating the IcINN multiple times. This involves different initializations of the network, while using exactly the same MC and data inputs. As we can see in Fig. 5, these fluctuations lead to relatively small variances. The corresponding correlations can be either positive or negative (see Fig. 6, first row).

Next we consider the uncertainties originating from statistical fluctuations in data and MC, which are dominant in our example. It is worth noting that all these statistical uncertainties of the cINN-based methods are amplified with the increasing number of iterations (see Fig. 5), similarly to what one observes for matrix-based methods.

For iterative (matrix- or cINN-based) unfolding procedures, the data component of the statistical covariance matrix has only positive correlations at the first iteration (see Fig. 6, 2nd row left).

This can be easily understood in the binned approaches: if e.g. one bin at the reconstructed-level fluctuates up (down) statistically, this will induce upward (downward) fluctuations in all the bins of the unfolded distribution. That's because the migration probabilities towards every bin are positive. A statistical fluctuation has a similar effect in the unbinned approaches, since the unfolded distribution for each unfolded event is positive-defined. This feature is indeed true up to the statistical fluctuations originating from the training and evaluation of the cINN, which can also be negative, as described above. One starts observing significant anti-correlations between nearby bins when using at least one reweighting of the Monte Carlo (see e.g. Fig. 6, 2nd row right). Indeed, this reweighting impacts the amount of migrations that the unfolding correction induces between different phase-space regions.

The MC component of the statistical covariance matrix does have anti-correlations, because the number of events in the unfolded distribution is preserved (i.e. equal to the one in the reconstructed-level data) when unfolding with any transfer matrix (e.g. with some statistical realization of the transfer matrix from a statistical fluctuation). Indeed, fluctuations of the transfer matrix effectively change the amount of migrations between the reconstructed-level bins, and hence the amplitude of the unfolding correction. In the unbinned approach, for each data event, the integral of the distribution unfolded for migration effects is equal to one,

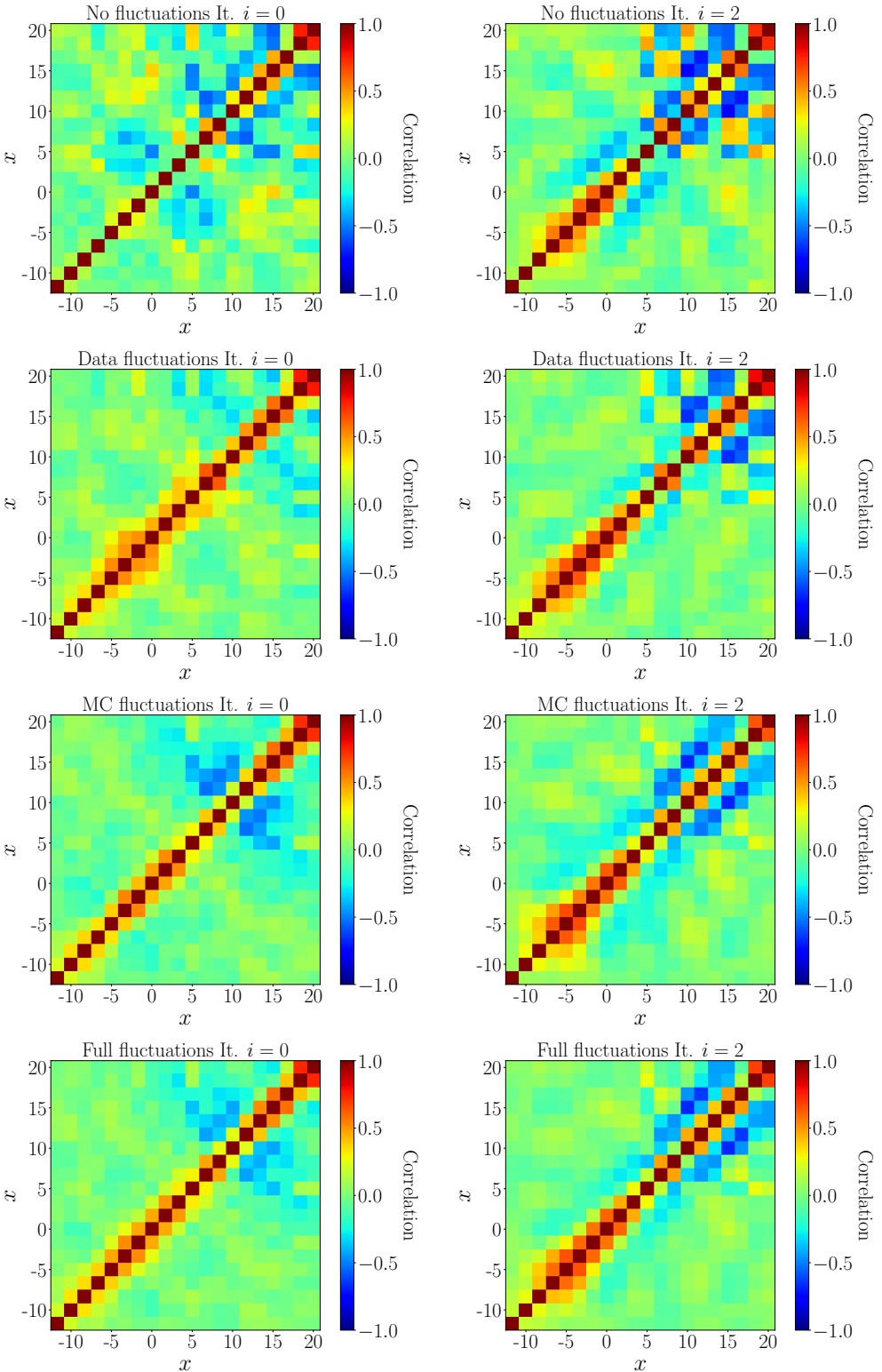

Figure 6: Correlation matrices of the IcINN unfolding results before reweighting (left) and after two reweightings (right), i.e. with $i = 0$ and $i = 2$ respectively, following the notations of Section 2.2. These correspond to the statistical uncertainties with no fluctuations for the input distributions (1st row), with fluctuations from data (2nd row), MC (3rd row) and total (4th row). When deriving these correlations, each event is unfolded 30 times and a number of $N_{\text{toys}} = 400$ bootstrap replicas is used.

while its shape is modified when fluctuating the weights of the input MC events. Therefore, these fluctuations induce anti-correlations between different phase-space regions covered by the unfolding result (see Fig. 6, 3rd row).

For some further use of the unfolded data, e.g. in re-interpretation studies, it is also relevant to derive the total statistical covariance matrix and the corresponding correlation matrix (see Fig. 6, 4th row), which naturally mixes the features of the data and MC components.

## 4  Unfolding $pp \to Z\gamma\gamma$ events

After the successful test on a toy model, we are now ready to apply the IcINN unfolding in a more realistic scenario, i.e. a process receiving significant contributions from the Standard Model (SM), while being also sensitive to potential contributions from beyond SM (BSM) physics. It is then possible to use the pure SM prediction as the Monte Carlo simulation on which we train our IcINN, while the (pseudo-)data that we unfold, receives a significant contribution from BSM physics. This means that the IcINN needs to unfold structures which are not part of the Monte Carlo simulation. While the presented example is low-dimensional, the approach expands naturally to higher dimensions as most network-based methods. The main training time is allocated for the initial training of the generative network, while subsequent trainings profit from the pre-trained model. The scaling behavior is hence expected to follow the same as standard generative network problems.

We use the Effective Field Theory (EFT) approach to parametrized the BSM contributions. The basic idea of EFT is to include additional terms into the SM Lagrangian which contain six- or eight-dimensional combinations of SM operators. The additional terms in the SMEFT Lagrangian implement new physics either via a characteristic change to a current interaction vertex or by introducing completely new interaction vertices. These operators are suppressed by a factor $\Lambda$, which is the scale where we expect new physics. For this study we choose the following process

$$pp \to Z\gamma\gamma\,, \qquad Z \to \mu^-\mu^+\,. \tag{20}$$

A leading order Feynman diagram for the SM contribution is shown in Fig. 7, left. For this process the anomalous triple gauge couplings (aTGC) introduced by the dimension-6 operators do not contribute. In addition, the anomalous quartic gauge couplings (aQGC) introduced via the dimension-6 operators do not contribute to this process as well, since they do not give rise to purely neutral aQGCs. A significant EFT contribution can hence be implemented using the dimension-8 extension

$$\mathcal{L}_{T,8} = \frac{C_{T,8}}{\Lambda^4} B_{\mu\nu}B^{\mu\nu}B_{\alpha\beta}B^{\alpha\beta}\,, \tag{21}$$

where we defined the Wilson coefficient $C_{T,8}$ as well as

$$B^{\mu\nu} = \partial^\mu B^\nu - \partial^\nu B^\mu\,, \tag{22}$$

with the generator $B^\mu$ associated with the $U(1)_Y$ gauge group of the weak hypercharge $Y$. $\mathcal{L}_{T,8}$ introduces aQGCs of the neutral electroweak gauge bosons: $ZZZZ$, $ZZZ\gamma$, $ZZ\gamma\gamma$, $Z\gamma\gamma\gamma$ and $\gamma\gamma\gamma\gamma$ [44]. The last three aQGCs enable additional Feynman diagrams at leading order, an example is displayed on the right in Fig. 7. We use MadGraph5 [45] and Pythia8.3 [46] for the event generation and DELPHES [47] to simulate detector effects. The MC events to train the IcINN are obtained using the regular SM simulation, for the pseudo-data we add the EFT contribution described above using the Eboli package [44]. The Wilson coefficient $C_{T,8}$ as well as the scale for new physics $\Lambda$ are collectively set to

$$\frac{C_{T,8}}{\Lambda^4} = \frac{2}{\text{TeV}^4}\,. \tag{23}$$

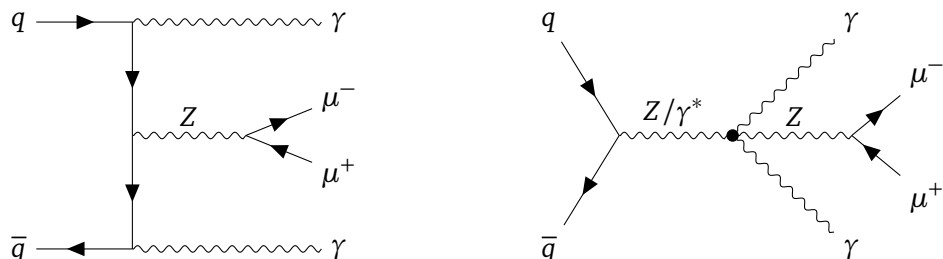

Figure 7: Feynman diagrams for the process $q\bar{q} \to Z\gamma\gamma$ with $Z$ decaying into muons. On the left the Standard Model leading order process is shown, on the right additional contributions that appear after including the EFT operator enabling the anomalous quartic gauge couplings $ZZ\gamma\gamma$ and $Z\gamma\gamma\gamma$.

This value has the same order of magnitude as current exclusion limits [48], the considered example being in this sense realistic. The observables that we are going to unfold are the transverse momenta of the muons, $p_T^-$ and $p_T^+$ for the negative and positive charge respectively, i.e. we are now performing an effective unfolding in two dimensions.

For the detector simulation we used the standard ATLAS-card provided by DELPHES in which, for simplicity we removed the rapidity-dependence of the muon momentum smearing.[3] The explicit momentum smearing is given as

$$\Delta p_T = p_T \cdot \sqrt{0.025^2 + 3.5 \cdot 10^{-8} \cdot \left(\frac{p_T}{\text{GeV}}\right)^2}.\tag{24}$$

In order to avoid covering too many orders of magnitude in our training data, we additionally implemented a high-$p_T$ cut at 250 GeV for the reconstructed muons. An alternative solution would consist in training several cINN's for various different $p_T$ ranges. The resulting distributions of $p_T^+$ and $p_T^-$, for the MC simulation and the pseudo-data, at both the truth and the reconstructed level, as well as their ratios, are shown in Fig. 8.

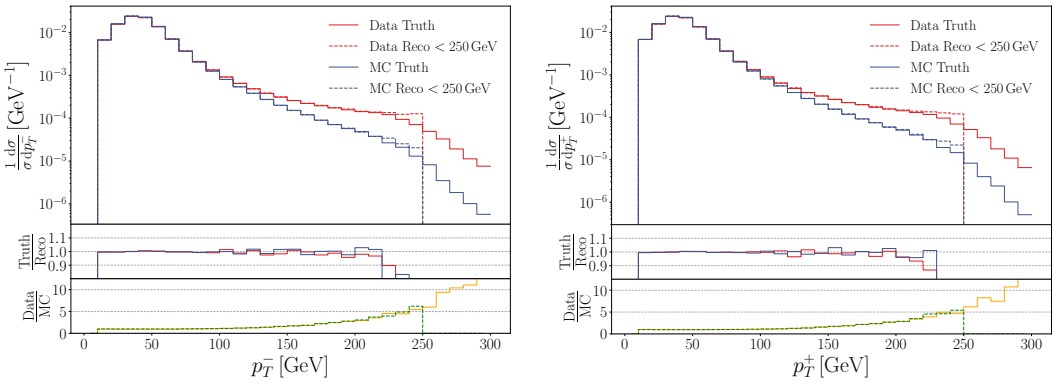

Figure 8: Simulated distributions of $p_T^-$ (left) and $p_T^+$ (right). The blue histograms are pure SM simulations and are used as the MC training events, while the red histograms contain additionally an EFT contribution and are used as pseudo-data. The continuous (dashed) lines indicate truth-level (reconstructed) quantities. In the lower parts of the plots the True/Reco and Data/MC ratios for the corresponding distributions are displayed.

---

[3]This modification is done in order to avoid hidden observables that have an impact on the detector smearing, while not being explicitly used in the IcINN. In an application of this algorithm to real data, the ultimate goal is to simultaneously unfold all observables measured by the detector, which will also avoid this kind of problems.

Following the same procedure as detailed in Section 2.2 we now perform the iterative trainings of the cINN and the classifier. Since we no longer have analytic predictions, we perform multiple closure tests to ensure the convergence of each training (see Appendix B). The hyperparameters of the corresponding cINN and the classifier architectures and training are given in Table 2 in the Appendix A.

Having validated the closure of the models, we can now consider the results of the unfolding obtained for various numbers of iterations, as displayed in Fig. 9. We notice that already before the first reweighting step, the cINN unfolded distribution shows a very good agreement with Data truth over a broad range of the phase space. The main deviations can be observed in the tail of the distribution, where the EFT contributions are largest. The comparison of the results obtained in the subsequent iterations shows a systematic improvement in these regions, progressively reducing the bias induced by the data-MC shape differences.

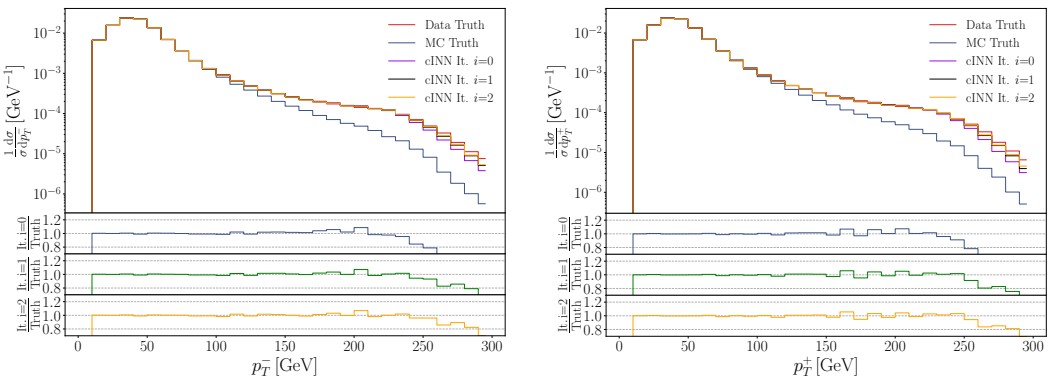

Figure 9: Unfolding results at iteration $i = 0$ (purple), iteration $i = 1$ (black) and iteration $i = 2$ (orange), compared with the truth distributions in data (red) and in MC (black), for $p_T^-$ (left) and $p_T^+$ (right). The bottom panels show the ratio between the unfolded results and the truth data distributions.

When unfolding multiple observables simultaneously, it is crucial to reproduce the correct correlations among them. The upper plot of Fig. 10 displays the result of the two-dimensional unfolding of $p_T^-$ and $p_T^+$. In the lower row we show the relative deviation when comparing this two-dimensional distribution with the detector- and the truth-level data ones. While the procedure preserves the correlations between the two unfolded quantities, the residual differences w.r.t. the desired result are typically smaller than the unfolding corrections themselves.

Finally, Fig. 11 shows the impact of the iterative unfolding on a single data event, for both the $p_T^-$ and $p_T^+$, over three iterations. While at low-$p_T$ the unfolding result is stable when increasing the number of iterations, in the high-$p_T$ region a clear downscaling (upscaling) of the lower (higher) bins is observed. This dependence of the per-event unfolding result on the number of iterations is coherent with the one displayed for the full distributions in Fig. 9.

# 5 Conclusion

In this paper we have presented the new iterative cINN-based unfolding algorithm IcINN. Starting from the cINN based unfolding, it progressively reduces the data-MC distribution shape differences, minimizing the impact of the MC simulations for the unfolded quantities. At the same time the IcINN preserves the ability of the cINN to unfold reconstructed quantities from individual data events to a (multidimensional) probability distribution at truth level. This enables the possibility of performing an event-by-event unfolding of experimental data for

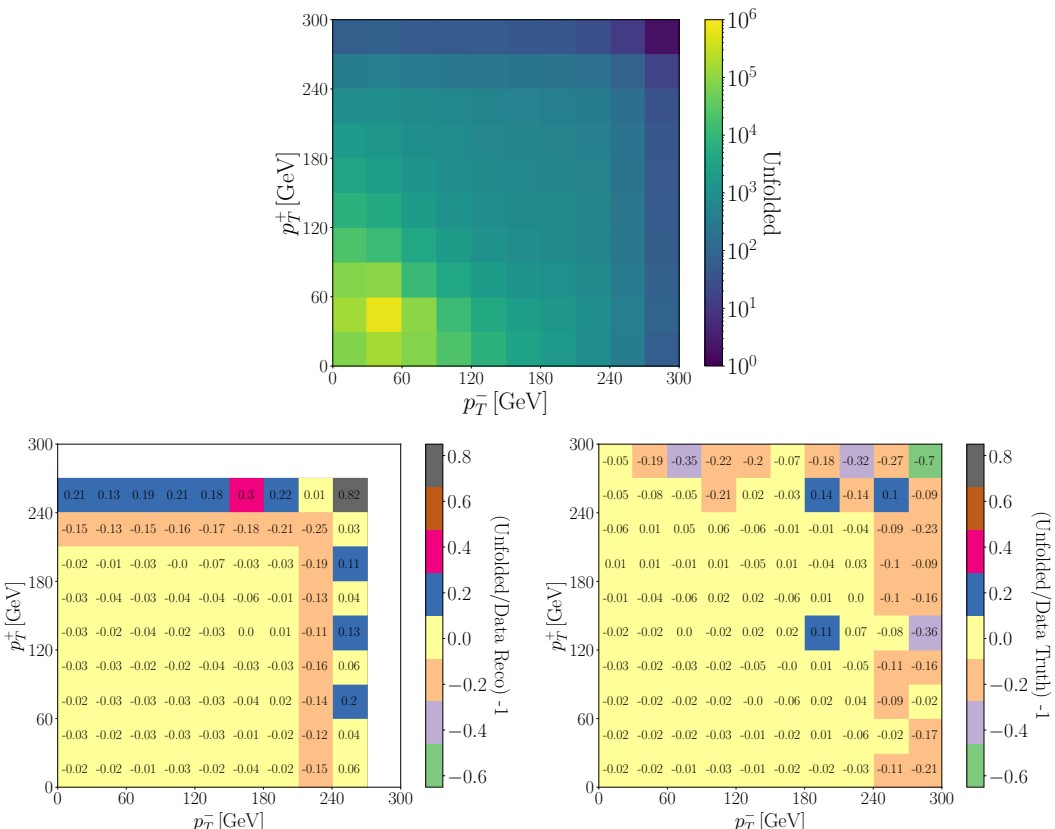

Figure 10: The upper row shows the matrix of the unfolded distribution after iteration $i = 2$ of $p_T^-$ and $p_T^+$. The lower row shows the ratios of this unfolded distribution w.r.t. the detector-level (left) and truth-level (right) data.

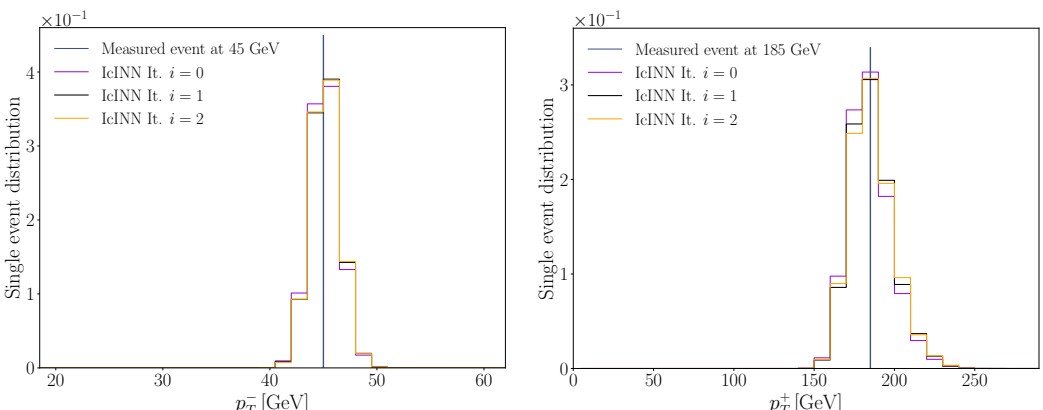

Figure 11: Single-event unfolding exemplified for an event with $p_T^- = 45$ GeV (left) and $p_T^+ = 185$ GeV respectively. The reconstructed value and the unfolded distributions over three iterations are displayed. In the left plot the bin size is decreased with respect to the standard binning used in this example to visualize the distribution.

numerous observables simultaneously, which is an important advantage compared to matrix-based methods, while also mitigating possible biases related to data-MC distribution shape differences.

We have demonstrated the reliability of the algorithm on a fully controllable toy model and tested it on a representative example for applications with real data. Finally we studied the

induced statistical uncertainties and the corresponding correlations among different phase-space regions. These represent a crucial aspect for real data measurements and subsequent phenomenological applications.

In order to enable the use of IcINN in the future, the python code as well as an instructive toy example have been prepared in a repository which can be made available upon request [49].

# Acknowledgments

MB would like to thank Philipp Ott for multiple helpful discussions about the $Z\gamma\gamma$ simulation.

**Funding information**  AB would like to acknowledge support by the BMBF for the AI junior group 01IS22079. AB and BM gratefully acknowledge the continuous support from LPNHE, CNRS/IN2P3, Sorbonne Université and Université de Paris.

# A  IcINN parameters

The Table 1 provides the list of parameters choosen for the cINN and the classifier, when applied for the toy example, while Table 2 provides the similar information for the $Z\gamma\gamma$ study. The training of the classifier is performed on the MC truth and the unfolded data reco distribution. We unfold each event one single time. In a more involved analysis it is possible to unfold each data reco event multiple times. In this case we can either take into account the multiplication factor as an additional weight in the loss function of the classifier or compensate the imbalance of MC truth vs unfolded events by generating more MC events.

Table 1: Parameter choice for the cINN and the classifier for the toy example. We used a one cyclic learning rate as well as the ADAM optimizer with a standard parametrization in both networks. The unfolded events of the cINN which were used to train the classifier were obtained by unfolding 3 times each of the 400 000 events (generated through a random sampling of the truth Gaussian distribution, followed by a random smearing according to the resolution function to obtain the reconstructed quantity).

| Parameter | cINN | Classifier |
|---|---|---|
| Conditional coupling blocks | 5 | - |
| Layers (per block) | 2 | 3 |
| Units (per layer) | 32 | 8 |
| Epochs | 100 | 100 |
| Learning rate | $10^{-4}$ | $10^{-3}$ |
| Maximum learning rate | $3 \cdot 10^{-4}$ | $3 \cdot 10^{-3}$ |
| Weight decay | 0.01 | - |
| Batch size | 4096 | 4096 |
| Number of training events | 500 000 | 1 200 000 |

Table 2: Parameter choice for the cINN and the classifier for the $Z\gamma\gamma$ dataset. We used a one cyclic learning rate as well as the ADAM optimizer with a standard parametrization in both networks. The unfolded events of the cINN which were used to train the classifier were obtained by unfolding 1 500 000 events once.

| Parameter | cINN | Classifier |
|---|---|---|
| Conditional coupling blocks | 5 | - |
| Layers (per block) | 4 | 6 |
| Units (per layer) | 64 | 32 |
| Epochs | 100 | 100 |
| Learning rate | $10^{-4}$ | $10^{-3}$ |
| Maximum learning rate | $3 \cdot 10^{-4}$ | $3 \cdot 10^{-3}$ |
| Weight decay | 0.01 | - |
| Batch size | 4096 | 4096 |
| Number of training events | 1 500 000 | 1 500 000 |

# B Closure checks for the iterative unfolding algorithms

Fig. 12 shows a sanity closure check, comparing the result of unfolding the reweighted reconstructed MC with the reweighted truth MC, at the various iteration steps. Good agreement is observed at all the iteration steps, indicating a reliable training of the cINN. The closure check for data, i.e. the comparison between the unfolded data and the corresponding truth distribution, also indicates an improvement with the increasing number of iterations. The closure check for MC shows much smaller deviations than the one performed for data, as expected, since only the latter is impacted by data-MC differences. Fig. 13 shows a check of the classifier, comparing the reweighted truth MC and the unfolded data distributions, at various iteration steps. Here also, good agreement is observed.

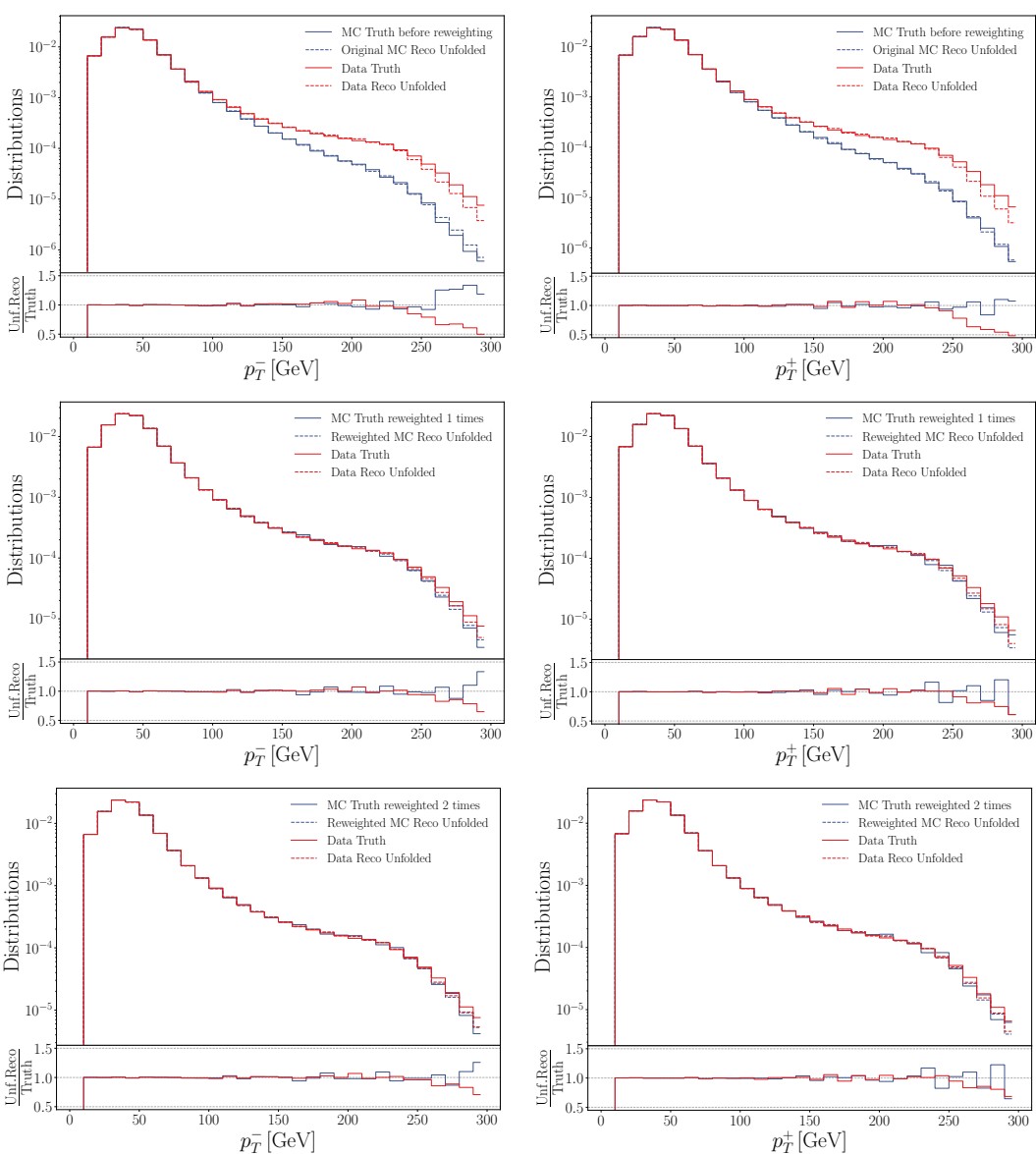

Figure 12: Result of the unfolding of the reweighted reconstructed MC (dashed blue) compared with the reweighted truth MC (blue line) at the first (top), second (middle) and third (bottom) iterations, i.e. after 0, 1 or respectively 2 reweightings, for $p_T^-$ (left) and $p_T^+$ (right). The data truth shape (red line) and the result of its unfolding (dashed red), at various iteration steps, are also indicated. The bottom pannels indicate the ratios of the unfolding results and truth distributions, for data and MC respectively.

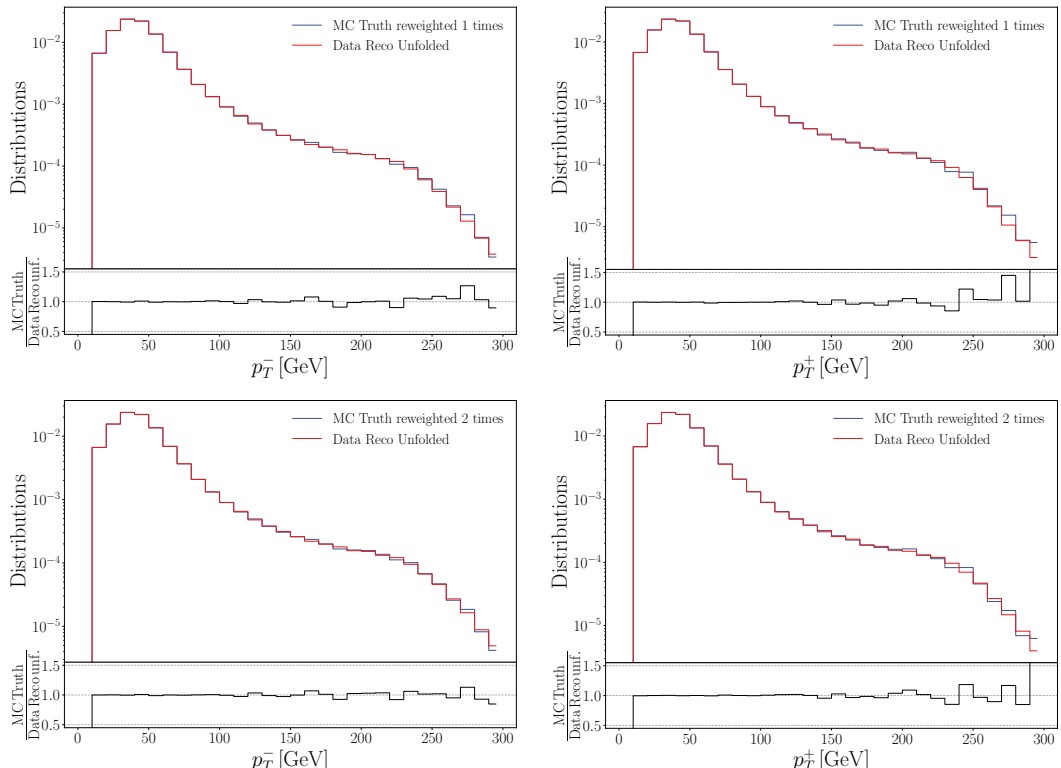

Figure 13: Classifier check, comparing the reweighted truth MC (blue) and the unfolded data distributions (red), after reweighting once (top) and after reweighting twice (bottom), for $p_T^-$ (left) and $p_T^+$ (right).

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
