# Peer review of "An unfolding method based on conditional Invertible Neural Networks (cINN) using iterative training"

_SciPost Physics, doi:SciPost Phys. Core 7, 007 (2024)_

## Round 2 · Referee Report · Anonymous (Referee 1) · 2023-8-16

Strengths

1- Novel use of the conditional Invertible Neural Networks (cINN) method, which is well-regarded in other disciplines. 2- Well written, well presented.

Weaknesses

1- The paper showcases the cINN method primarily through simple, low-dimensional examples. More examples would be helpful. 2- There is a gap in the paper, as there is no comparative study against contemporary unfolding methods, making it challenging to ascertain its advantages.

Report

The study first introduces the collider data unfolding problem, as well as the prevalent methodologies used to tackle it, underscoring their limitations. The second section proceeds to redefine the problem through Bayesian framework. Within this framework, the manuscript introduces the cINN unfolding algorithm and the iterative approach involving training-unfolding-reweighting. The authors then put their method to the test in two distinct scenarios. The initial test (third section) involves a simplistic 1D toy model utilizing a Gaussian distribution. The subsequent scenario revolves around a more complex pp->Z\gamma\gamma process.

In summary, while the manuscript presents a novel approach to unfolding detector data, certain areas could be refined to provide a more comprehensive and clearer picture of the methodology's potential and comparative advantages.

Requested changes

Here are some minor feedbacks, which would be good to addressed but not necessary.

1- Very few typos: a ... flow which have, reweigthing. Some minor British/American difference in word usage. 2- I would give ref to ADAM. 3- Figure (6) is hard to follow. To enhance clarity, adopting a 3-color scheme would be recommended. This would allow readers to distinguish between positive and negative correlations. 4- Section 4 parameter space truncates at a relatively low energy level. The manuscript could benefit from elaborating on training time/complexity for extended parameter spaces. 5- The concluding remarks of the manuscript seem rather concise. Expanding this section could provide a clearer picture of the manuscript's impacts. Specifically, it would be insightful to: 5.1- Dive into the potential of the method when applied to processes with higher dimensions, i.e. more complicated processes beyond two jets. 5.2- Pit the cINN approach against other methods. If a direct comparison isn't feasible, providing quantitative evidence highlighting its superior efficiency would be beneficial, e.g. quantifying the reduction in variance, how much bias is reduced in distribution.

  • validity: top
  • significance: top
  • originality: high
  • clarity: high
  • formatting: perfect
  • grammar: perfect

Author:  Anja Butter  on 2024-01-11  [id 4238]

(in reply to Report 1 on 2023-08-16)

Dear Referee,

Thank you very much for the positive feedback and your suggestions. We have addressed them in the following way:

1 -> We have corrected the typos. We try to write everything in American English now.

2 -> The reference to ADAM has been included.

3 -> We adapted a new color scheme which should visualize the correlations better.

4 -> We have added a comment on the expected training behavior towards higher dimensions.

5 -> In the conclusions of this paper the goal was mainly to summarize its content. We especially wanted to avoid putting somewhat speculative comments about more comparisons with other methods, which were left for later studies. The referee can actually find a follow-up study on such comparisons in a more recent preprint (arxiv:2310.17037). Still, we have slightly expanded the conclusions to better emphasize the benefits of the IcINN method and the way it was tested.

Anonymous on 2024-01-16  [id 4246]

(in reply to Anja Butter on 2024-01-11 [id 4238])

Dear authors,

We have happy with these changes and recommend the updated version (https://arxiv.org/pdf/2212.08674v3.pdf) for publication. Please go ahead and submit the revision.

Best regards,

---

## Editorial Decision

published